# Response of Common Garden Plant Leaf Traits to Air Pollution in Urban Parks of Suzhou City (China)

**Zhiyu Yang [1], Xing Zhang [1,2,*], Yanting Qu [2], Fei Gao [1] and Yutong Li [1]**

[1]  School of Architecture and Urban Planning, Suzhou University of Science and Technology, Suzhou 215129, China; yzzyworkin@163.com (Z.Y.); gaofei@usts.edu.cn (F.G.); zl_y98@foxmail.com (Y.L.)
[2]  Institute of Nature and Ecology, Heilongjiang Academy of Sciences, Harbin 150040, China; quyanting1976@126.com
[*]  Correspondence: 2605@usts.edu.cn

**Abstract:** In this study, to comprehensively investigate the impact of garden plants on air quality, we measured the leaves of 18 common garden plants in three different areas, namely, Suzhou industrial parks (clean air area (CAA)), Xiangcheng district parks (lightly polluted area (LPA)), and Huqiu district parks (highly polluted area (HPA)). We also measured the leaf functional traits of different life-types of plants. To explore the trade-off strategies of the leaf traits of common garden plants in response to air pollution and to assess the adaptive capacity of different life types of plants. The results show that plants in the polluted area had higher leaf dry matter content (LDMC) and leaf nitrogen content per unit mass ($N_{mass}$), and a lower specific leaf area (SLA), maximum net photosynthetic rate per unit area ($A_{area}$), transpiration rate (Tr), stomatal conductance (Gs), and chlorophyll value (SPAD). Pearson correlation analysis showed that SLA was significantly positively correlated with $N_{mass}$, Tr, photosynthetic use efficiency (PNUE), and SPAD, and significantly negatively correlated with LDMC, while $A_{area}$ was significantly positively correlated with chlorophyll value. Redundancy analysis revealed that the correlation between each leaf functional trait and atmospheric pollution factors was as follows: LDMC > $N_{mass}$ > SLA > LA > $A_{area}$ > Tr > PNUE > SPAD. The results suggest that different plant types have varying levels of adaptability to environmental conditions. Trees were found to be the most adaptable, followed by shrubs, herbs, and lianas. Additionally, under the stress of air pollution, herbs and lianas exhibited characteristics of "fast investment-return" on the leaf economic spectrum, meaning they were able to quickly allocate resources to maximize their return. However, trees and shrubs displayed traits of "slow investment-return", indicating a more conservative approach to resource allocation. These results provide valuable insights into the leaf trade-off strategies of plants in Suzhou Park under air pollution stress and can guide the selection of suitable plant species in similar environments.

**Keywords:** garden plants; leaf economic spectrum; air pollution; leaf functional traits; trade-off strategies

## 1. Introduction

Since 1978, China has undergone rapid development, resulting in the significant emission of air pollutants that pose a threat to human respiratory health and physiological function, hinder urban ecological construction, and impede the healthy development of cities [1]. According to air pollution data from 2018, only one-third of the 338 monitored cities in China met the air quality standards [2,3]. Landscape plants play a crucial role in urban vegetation communities, providing diverse forms and species that facilitate the creation of aesthetically pleasing urban environments [4]. In addition to their ornamental function, plants have ecological functions, including carbon sequestration, oxygen release [5–7], dust retention, and noise reduction [8–10]. The leaves of plants serve as the primary site for energy conversion and material exchange with the external environment,

and their functional traits can respond sensitively to environmental changes. Wright's Leaf economics spectrum (LES) theory, first proposed in 2004 [11], suggests that plants adopt a strategy of acquisition or conservatism, depending on the availability of resources, and trade-off resource allocation to enhance their adaptive capacity under environmental stress [12,13]. The LES theory offers a quantitative framework for predicting plant resource utilization efficiency and trade-off strategies [14]. However, it is crucial to validate the accuracy of the LES theory at the local scale [15]. Extensive research has demonstrated that leaf economic spectra are primarily influenced by environmental factors such as light, temperature, moisture, and notably, atmospheric pollutants including $CO_2$, $CO$, $SO_2$, $NO_2$, and $O_3$. Atmospheric pollution emerges as a significant factor impacting plant growth and its application in landscaping. The response of the leaf economic spectrum to atmospheric pollution predominantly centers on leaf structural and chemical traits. Osnas et al. have highlighted that specific leaf area (SLA) represents one of the trade-offs adopted by plants when faced with limited environmental resources [16]. In the presence of atmospheric pollution, plants may modify their resource allocation strategy, directing more resources towards growth and production capacity. This adjustment aims to fortify defensive tissues and structures, thereby enhancing vitality and resistance [17,18]. Chen and his team conducted a study on the leaf functional traits of 89 plant species in the eastern part of Guangdong Province, China [19]. Their findings revealed significant differences in specific leaf area (SLA) and leaf dry matter content (LDMC) among the different species. Under the stress of atmospheric pollution, the plants exhibited a tendency towards low SLA and high LDMC [20]. The response of herbaceous plants to $O_3$ was studied by Evans et al., who found no significant differences in leaf thickness, fenestrated tissue thickness, and spongy tissue thickness [21]. Other studies have reported an increase in the thickness of epidermal tissues and a decrease in the thickness of chloroplastic tissues in acacia leaves contaminated with $NO_2$ and $SO_2$ [22]. Hang and other scholars demonstrated that environmental stresses, such as vehicle exhaust and soot, resulted in a decreasing trend in leaf nitrogen content of garden shrubs in South China [23]. Additionally, Lai conducted a study on the leaf functional traits of garden plants under different concentrations of $NO_2$ fumigation, and the results showed significant differences in the nitrogen and phosphorus contents of the leaves of different plants in response to varying concentrations of $NO_2$.

Chlorophyll is an important pigment for photosynthesis in green plants, which not only absorbs and converts light energy, but also its content directly affects the photosynthetic capacity of green plants [24]. At the same time, chlorophyll is also one of the most important indicators of the degree of external stress on plants [25,26]. Different types of pollution in the ecological environment will affect the chlorophyll content of green plants, thus reflecting the level of plant resistance and sensitivity. Therefore, the change of chlorophyll content can be used as one of the important indicators to assess the ecological adaptability of plants. Gao Chuanyou et al. [27] showed that when plants are exposed to pollution stress for a long period of time, their photosynthetic efficiency decreases, resulting in a lack of basic substances required for growth and development. This is due to the effect of pollution on plants, resulting in the decomposition of chlorophyll a and b, which leads to a decrease in the total chlorophyll content [28]. Plant photosynthesis is also extremely sensitive to air pollution, and studies have shown that the exposure of plants to bursts of high $SO_2$ concentrations in the vicinity of plants can lead to a cessation of photosynthesis. Different species of plants respond differently to $SO_2$ concentrations. In a study by Samuel B M et al. [29], pine, spruce, larch and lime were fumigated with $SO_2$ for one hour. The results showed that photosynthesis was reduced in all four species. For crops such as soybean, wheat, rice and potato, Wu Liying et al. [30] conducted $SO_2$ fumigation tests and found that their photosynthesis was inhibited to different degrees.

In addition, there are many studies focusing on the effect of plant photosynthetic traits on $NO_2$. Okano K [31] showed that the net photosynthetic rate of gerbera increased after two weeks of $NO_2$ fumigation at a concentration of 0.2 μL/L, but $NO_2$ at concentrations of 0.5 μL/L and 1.0 μL/L inhibited the net photosynthetic rate of the plant to varying

degrees. Sabaratnam and Gupta [32,33] found that the net photosynthetic rate of soybean was significantly reduced at 0.4 µL/L of $NO_2$ and increased at 0.2 µL/L of $NO_2$. The current study mainly focuses on the effects of atmospheric pollutants on the photosynthetic characteristics of plants, mainly related to $SO_2$, $NO_2$ and other aspects, but the study of the comprehensive effects of atmospheric pollutants is still limited. These studies highlight the sensitivity of leaf functional traits to atmospheric pollution and the potential for plants to adapt to environmental stress through adjustments in their leaf anatomy and nutrient allocation strategies [34].

In recent years, Chinese scholars have conducted extensive research on the Leaf economics spectrum (LES) theory, investigating various factors such as altitude [35], thermal environment [36], light intensity [37], and moisture gradients [38]. Specifically, studies conducted in China have explored the response of leaf functional traits to atmospheric pollution. Zhu [39] and Li [40] conducted experiments on greening tree species in Beijing and street trees in Suzhou, respectively, and observed the presence of a global leaf economic spectrum even in the presence of atmospheric pollution.

The Leaf economic spectrum (LES) theory posits that plants employ diverse resource allocation strategies in response to resource availability, thereby enhancing their ability to adapt to environmental stress. Leaf functional traits, including leaf dry matter content (LDMC), specific leaf area (SLA), leaf nitrogen content per unit mass ($N_{mass}$), and maximum net photosynthetic rate per unit area ($A_{area}$), play a vital role in this trade-off strategy. In this paper, the following leaf functional traits were selected as indicators for analysis: leaf dry matter content (LDMC); specific leaf area (SLA); the net photosynthetic rate per unit area ($A_{area}$); stomatal conductance (Gs); transpiration rate (Tr); the nitrogen content of leaves per unit mass ($N_{mass}$); the calculation of photosynthetic nitrogen utilization (PNUE); and chlorophyll values (SPAD).

This study aims to investigate the key issues in the planning and design of urban park plant landscapes. Based on the leaf economic spectrum traits, the study aims to scientifically evaluate the resource trade-off strategies of different life types and species of plants under atmospheric changes, and to investigate the responses of common garden plants under different environmental gradients, so as to provide new ideas and reference bases for the design of landscape plants.

## 2. Materials and Methods

### 2.1. Study Area

Suzhou, Jiangsu Province (119°55′~121°20′ E, 30°47′~32°02′ N) is located on the eastern coast of mainland China, with a subtropical monsoon maritime climate, four distinct seasons, and an annual rainfall of about 1100 mm. The average annual temperature is 15.7 °C. Suzhou is located in the intersection of northern subtropical and central subtropical zones. The bioclimatic characteristics, affected by the southeast monsoon, presenting a warm and humid climate. The natural vegetation belongs to the northern subtropical deciduous, evergreen broad-leaved mixed forest zone. Suzhou, located in the Yangtze River Delta city cluster, boasts a warm and humid climate along with abundant rainfall. It is renowned for its historical and cultural significance, showcasing precious classical gardens and landscapes. The city has a rich tradition of carefully selecting and matching landscape vegetation.

The city's economic structure is dominated by industry, and the industrial structure is dominated by heavy industry and mostly located in the upwind direction of the city. The number of key industrial enterprises in Suzhou reaches 10,233, contributing to 96.7% of the $SO_2$, 59.9% of the $NO_x$, and 49.7% of the $PM_{2.5}$.

In terms of time distribution, the $PM_{2.5}$ exceedance pollution is most serious in autumn and winter, and the $PM_{2.5}$ exceedance pollution is least serious in July to September. $PM_{2.5}$ mainly comes from the local area of Suzhou, mainly from motor vehicle exhaust, which reaches 33.6%, followed by coal combustion with 26.0%, industrial process with 12.8%, dust with 12.7%, biomass combustion with 5.9%, and other emission sources with 8.1%.The

emissions of other pollutant are as follows: $SO_2$—98,400 tonnes; $No_x$—21,200 tonnes; $PM_{2.5}$—72,700 tonnes; and $PM_{10}$—156,600 tonnes. The main sources of $SO_2$ are electricity, iron and steel, and textile; the main sources of $No_x$ are electricity, iron and steel, and motor vehicles; and the main sources of $PM_{2.5}$ and $PM_{10}$ are iron and steel, road dust, construction dust, and electricity.

According to monitoring data requested by the research team from the Meteorological Service of the City, three administrative districts in Suzhou with different air pollution levels were selected (Figure 1), namely, Xiangcheng district (lightly polluted area (LPA)), Hi-tech district (highly polluted area (HPA)) in the heavily polluted area and Suzhou Industrial Park (clean air area (CAA)) in the clean air area as the control, and three parks in each area were selected for sampling experiments. The air quality data in this study were obtained submitting a request to the official website of Suzhou Bureau of Ecology and Environment (suzhou.gov.cn), and the data included the daily average values of $SO_2$, $NO_2$, $PM_{10}$, $PM_{2.5}$, and the daily average value of the air quality index (AQI) from 1 October to 30 November 2021.

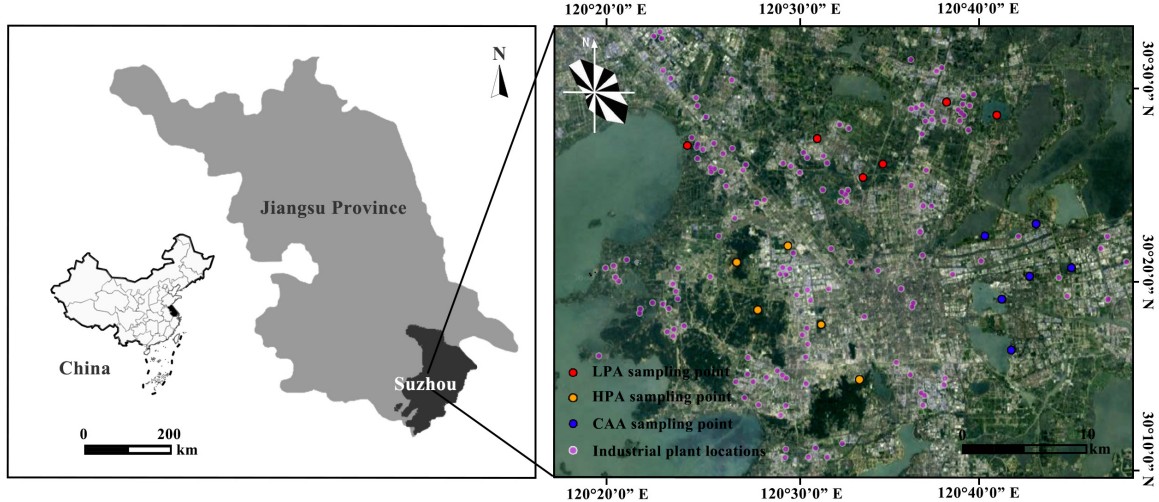

**Figure 1.** Study area and gradient zone delineation.

### 2.2. Collection and Measurement of Samples

The sampling areas in this paper are all urban parks and green spaces with good soil homogeneity, which ensures the relative consistency of tree species and management conditions. Sampling selected 18 species of similar age, good growth, and that are representative of the common garden tree species in Suzhou City according to the life type and are divided into four categories: trees, shrubs, herbs and lianas. The study of the sub-four categories of tree species life types in the different air quality environment, response, and trade-off mechanisms is shown in Table 1.

### 2.3. Sample Collection and Measurement

Plants with high rainfall in spring and summer have a higher capacity to settle and adsorb pollutants due to their high metabolic activity [41], and the pollutants accumulated by the plants reach their peak in autumn [42,43].

Therefore, the present study was conducted in autumn (early October to late November), with a total of 32 days of sampling, from 9:00 a.m. to 15:00 p.m., and no less than 30 plants of each tree species were collected, with 20 leaves per plant. The sampling sites were clusters of trees, and the park management office was consulted before sampling to ensure that the plants were of similar age and in good condition. We selected trees of similar diameter at breast height for sampling. Mature fresh leaves were collected to reflect the growth of the plant, free from pests and diseases, and then placed in Ziplock bags and refrigerated in the laboratory for later use.

**Table 1.** Eighteen experimental plant species in Suzhou.

| Number | Plant-Life | Species | Familia | Genus |
|---|---|---|---|---|
| 1 | Tree | *Cinnamomum camphora* (L.) J.Pres | *Lauraceae* | *Camphora* |
| 2 | | *Osmanthus fragrans* (Thunb.) Lour | *Oleaceae* | *Osmanthus* |
| 3 | | *Magnolia grandiflora* L. | *Magnoliaceae* | *Magnolia* |
| 4 | | *Koelreuteria paniculata* Laxm. | *Sapindaceae* | *Koelreuteria* |
| 5 | | *Ginkgo biloba* L. | *Ginkgoaceae* | *Ginkgo* |
| 6 | | *Sapindus saponaria* L. | *Sapindaceae* | *Sapindus* |
| 7 | Shrub | *Loropetalum chinense* (R. Br.) Oliver | *Hamamelidaceae* | *Loropetalum* |
| 8 | | *Viburnum odoratissimum* Ker Gawl. | *Viburnaceae* | *Viburnum* |
| 9 | | *Pittosporum tobira* (Thunb.) W. T. Aiton | *Pittosporaceae* | *Pittosporum* |
| 10 | | *Hibiscus mutabilis* L. | *Malvaceae* | *Hibiscus* |
| 11 | | *Buxus sinica* (Rehder & E. H. Wilson) M. Cheng | *Buxaceae* | *Buxus* |
| 12 | | *Lagerstroemia indica* L. | *Lythraceae* | *Lagerstroemia* |
| 13 | Herb | *Ophiopogon bodinieri* H. Lév. | *Asparagaceae* | *Ophiopogon* |
| 14 | | *Oxalis corniculata* L. | *Oxalidaceae* | *Oxalis* |
| 15 | | *Ophiopogon japonicus* (L. f.) Ker Gawl. | *Asparagaceae* | *Ophiopogon* |
| 16 | Liane | *Jasminum mesnyi* Hance | *Oleaceae* | *Jasminum* |
| 17 | | *Parthenocissus tricuspidata* (Siebold & Zucc.) Planch. | *Vitaceae* | *Parthenocissus* |
| 18 | | *Trachelospermum jasminoides* (Lindl.) Lem. | *Apocynaceae* | *Trachelospermum* |

### 2.3.1. Measurement of Leaf Structural Traits

The cleaned leaves were placed on an analytical balance to weigh the leaf fresh weight (LFW) and vernier calipers to measure the leaf thickness (LT), with 30 slices measured for each species. The leaves were first placed in an oven at 80 °C for 3 h and then set to dry at 55 °C for 48 h to a constant weight. After removal, the leaves corresponding to the leaf fresh weight number were weighed again for leaf dry weight (LDW), and leaf dry matter content (LDMC) was calculated with the following Formula (1):

$$LDMC\left(g{\cdot}g^{-1}\right) = LDW(g)/LFW(g) \tag{1}$$

Leaf area measurement (LA), leaf length (LL), and leaf width (LW) were determined using a leaf area scanner (MICROTEK ScanMaker i800plus). Leaf mass per area (LMA) was calculated according to Formula (2), and specific leaf area (SLA) was calculated according to Formula (3).

$$LMA\left(g{\cdot}cm^{-2}\right) = LDW(g)/LA\left(cm^2\right) \tag{2}$$

$$SLA\left(cm^2{\cdot}g^{-1}\right) = LA\left(cm^2\right)/LDW(g) \tag{3}$$

Dried samples were ground and crushed through a sieve, and 1.0–15.0 mg of each sample was weighed using an analytical balance (XPE105, METTLER TOLEDO). Samples were wrapped in tin capsules and placed in an elemental analyzer (EURO EA3000) for the determination of the nitrogen content of leaves per unit mass ($N_{mass}$), and the mean value of the same species was taken as the value of the assay.

### 2.3.2. Measurement of Leaf Physiological Traits

Chlorophyll values (SPAD) of mature and undamaged plant leaves were determined using a portable chlorophyll meter (TYS-B) in an outdoor environment with natural light. The 3rd to 5th mature leaves of the plant at the front of the branches in the mid-height part of the sunny side were measured using a photosynthesizer (PPsystem, Li-Cor6400XT, Beijing, China), and the values were obtained in a standard leaf chamber. The instrument was set with LED red and blue light sources, light intensity of 1500 µmol m$^{-2}{\cdot}$s$^{-1}$, $CO_2$ concentration of 400 µmol L$^{-1}$, and leaf chamber temperature of 30 °C, and the average value was taken after three repetitions of measurement. The net photosynthetic rate per

unit area ($A_{area}$), transpiration rate (Tr), and stomatal conductance (Gs) were measured. The calculation of photosynthetic nitrogen utilization (PNUE) was performed using Formula (4):

$$\text{PNUE}\left(\mu mol \cdot g^{-1}N \cdot s^{-1}\right) = A_{area}\left(\mu molCO_2 m^2 \cdot s^{-1}\right)/N_{area}\left(g \cdot cm^{-2}\right) \tag{4}$$

*2.4. Data Processing and Statistics*

For leaf functional traits, SPSS 22.0 software was used, and the degree of variation in each of the leaf economic spectrum traits was assessed using the coefficient of variation. The calculation of the coefficient of variation (C.V) was performed using Formula (5):

$$\text{C.V} = (\text{SD}/\text{MN}) \cdot 100\% \tag{5}$$

where SD is the standard deviation, and MN is the mean.

One-way analysis of variance (ANOVA) was adopted to investigate the response of structural and physiological traits to different gradients of atmospheric pollution; Pearson correlation analysis (PCA) was used to investigate the correlation of leaf traits between different types of landscape plants and different gradients of atmospheric pollution; Canoco 5.0 software was used to carry out the redundancy analysis (RDA) was performed using Canoco 5.0 software to investigate the correlation between leaf traits of different landscape plants and atmospheric pollutants, and to screen the leaf traits related to air pollution.

To calculate the value of fuzzy membership function analysis (FMFA), when leaf traits are positively correlated with thermal environment, Formula (6) was used and is as follows:

$$R(X_i) = (X_i - X_{min})/(X_{max} - X_{min}) \tag{6}$$

If negatively correlated, Formula (6) is as follows:

$$R(X_i)_{inverse} = 1 - (X_i - X_{min})/(X_{max} - X_{min}) \tag{7}$$

where $X_i$ is the measured value of the i-th indicator, $X_{max}$ indicates the maximum value of the measured indicator, and $X_{min}$ is the minimum value of the measured indicator.

## 3. Results

*3.1. Air Pollution Gradient Analysis*

From the data of the Suzhou Eco-Environmental Bureau, the air pollution level in the three gradient zones follows the trend HPA > LPA > CAA. Figure 2 shows that compared to CAA, the content of $SO_2$, $NO_2$, $PM_{2.5}$, and $PM_{10}$ in HPA and LPA increased significantly, and that the urban air quality composite index AQI was the most distinctive among the three zones. The degree of air quality differentiation among the three gradient zones was significant. The range of contaminants is shown in Table 2.

**Table 2.** Atmospheric pollution concentration.

| Area | | $SO_2$ ($\mu g/m^3$) | $NO_2$ ($\mu g/m^3$) | $PM_{10}$ ($\mu g/m^3$) | $PM_{2.5}$ ($\mu g/m^3$) | AQI |
|---|---|---|---|---|---|---|
| CAA | Average value | 4.87 | 27.00 | 54.00 | 11.00 | 46.00 |
| | Min | 1.00 | 5.00 | 8.00 | 3.00 | 18.00 |
| | Max | 17.00 | 84.00 | 94.00 | 63.00 | 170.00 |
| LPA | Average value | 7.93 | 28.90 | 62.20 | 25.00 | 79.00 |
| | Min | 3.00 | 4.00 | 10.00 | 4.00 | 18.00 |
| | Max | 15.00 | 105.00 | 99.00 | 66.00 | 169.00 |
| HPA | Average value | 9.37 | 43.80 | 65.53 | 27.37 | 126.00 |
| | Min | 6.00 | 6.00 | 7.00 | 5.00 | 15.00 |
| | Max | 32.00 | 70.00 | 164.00 | 77.00 | 500.00 |

Note: The pollutant values in the table are 24 h average concentrations.

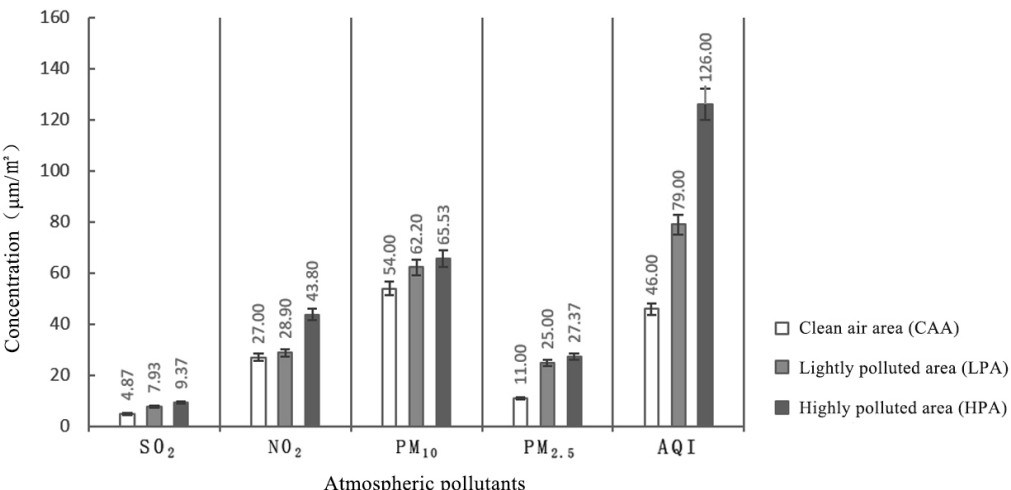

**Figure 2.** Air pollution.

*3.2. Relationships between Air Pollutants and Leaf Functional Traits*

In this study, redundancy analysis was used in order to investigate the relationship between air pollution factors and different leaf functional traits of 18 garden plants in Suzhou. Redundancy analyses were performed with plant leaf functional traits as species variable groups and each pollutant in the air environment as environmental variable groups. Each arrow points to the direction of the sharpest increase in the value of the corresponding environmental variable. The angle between the arrows indicates the correlation between the environmental variable and the leaf trait: a positive correlation is present when the angle is less than 90°, and a negative correlation is present when the angle is greater than 90°. The length of the arrows indicates the fit of the environmental variables.

The environmental correlations between each trait and air pollution were as follows: LDMC > $N_{mass}$ > SLA > LA > $A_{area}$ > TR > PNUE > SPAD. Each pollutant was positively correlated with the LDMC and $N_{mass}$ of trees, and negatively correlated with SLA, Gs, Tr, SPAD, $A_{area}$, and PNUE, with Axis 1 and Axis 2 explaining 76.17% and 4.82% of all the information, respectively, cumulatively explaining 80.99% of the information (Figure 3a). Each pollution factor was positively correlated with shrub LDMC, $N_{mass}$, and SLA, and negatively correlated with Gs, Tr, SPAD, $A_{area}$, and PNUE, with Axis 1 explaining 61.02% of all the information and Axis 2 explaining 1.48%, cumulatively explaining 62.5% of the information (Figure 3b). Vine LDMC, $N_{mass}$, and LA were positively correlated with each pollutant, and other leaf functional traits were negatively correlated with each pollutant factor, with Axis 1 explaining 82.96% of all the information and Axis 2 explaining 6.19%, with a cumulative total of 89.15% (Figure 3c). Herbaceous LDMC, $N_{mass}$, and LA were positively correlated with each air pollutant, and all other functional traits were negatively correlated, with Axis 1 and Axis 2 explaining 79.44% and 3% of all the information, respectively, and cumulatively explaining 82.44% of the information (Figure 3d). It can be seen that the first two axes clearly reflect the relationship between plant leaf functional traits and air pollutants, and are mainly determined by Axis 1, indicating that the degree of linear combination between the sorting axes and atmospheric environmental factors can better reflect the correlation between the environment and plant leaf functional traits, and the sorting results are reliable.

*3.3. Leaf Functional Traits*

ANOVA analysis was conducted on 18 garden tree species in Suzhou City, and it was concluded that the differences between the traits of different garden plant life type in the three air quality gradient zones in Suzhou City were significant (Table 3).

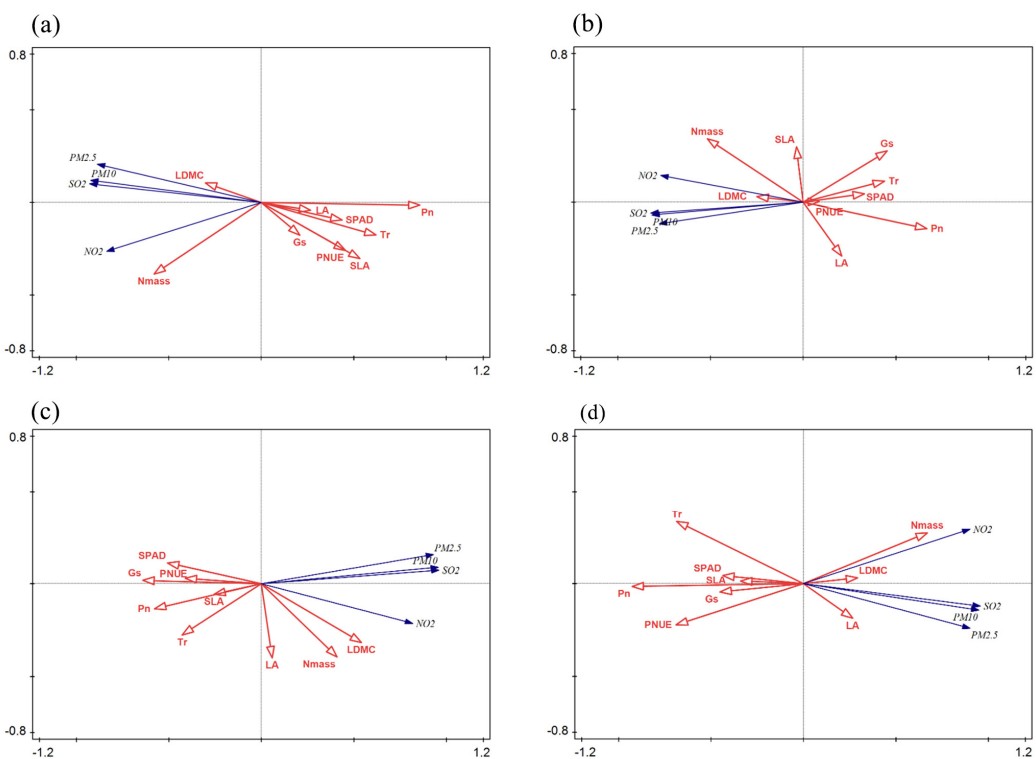

**Figure 3.** (**a**) Trees. (**b**) Shrubs. (**c**) Lianas. (**d**) Herbs.

In CAA and LPA, the LDMC was trees > shrubs > lianas > herbs, and in HPA, lianas > trees > shrubs > herbs; herb SLA was the largest in the three gradient zones, which was herbs > lianas > trees > shrubs in CAA and LPA, and herbs > lianas > shrubs > trees in HPA, and herbs were significantly different from trees, shrubs, and lianas. The $A_{area}$ of shrubs were the largest in all three gradient zones and differed significantly from other life types; Gs of different life types showed herbs > shrubs > lianas > trees in the clean zone and LPA, and shrubs > herbs > lianas > trees in the HPA; shrub Tr was the highest in all three gradient zones, and showed shrubs > trees > herbs > lianas in the CAA, shrubs > herbs > lianas > trees in the LPA, and shrubs > herbs > lianas > trees in the HPA. $N_{mass}$ was herbs > lianas > trees > shrubs in the CAA, lianas > herbs > trees > shrubs in the LPA, and lianas > herbs > shrubs > trees in the HPA; herb PNUE was the highest in all the three gradient zones, herbs > lianas > shrubs > trees in the CAA, and herbs > shrubs > lianas > trees in the remaining two zones; the SPAD of each plant life type was the highest in the CAA. In the CAA, the SPAD of each life type was lianas > shrubs > herbs > trees; in the LPA, it was lianas > shrubs > trees > herbs; and in the HPA, it was lianas > shrubs > tree s> herbs, and it formed a significant difference among the four life types.

**Table 3.** Plant leaf functional traits in gradient zones.

| Area | Plant-Life | LDMC | SLA | $A_{area}$ | Gs | Tr | $N_{mass}$ | PNUE | SPAD |
|------|-----------|------|-----|-------|-----|-----|-------|------|------|
| CAA | Tree | 0.433 ± 0.071 a | 10,013.975 ± 2606.067 b | 16.48 ± 3.705 b | 0.006 ± 0.004 a | 0.62 ± 0.952 a | 2.586 ± 0.395 b | 2.437 ± 1.148 b | 47.023 ± 9.302 a |
| | Shrub | 0.375 ± 0.047 a | 6659.786 ± 2683.895 b | 20.669 ± 2.898 a | 0.013 ± 0.008 ab | 0.762 ± 0.86 a | 2.058 ± 0.583 c | 8.012 ± 6.319 b | 51.42 ± 12.868 a |
| | Herb | 0.296 ± 0.092 b | 12,045.93 ± 7313.533 b | 18.767 ± 3.335 ab | 0.008 ± 0.002 bc | 0.27 ± 0.104 a | 2.68 ± 0.449 b | 17.039 ± 3.079 b | 52.994 ± 8.389 a |
| | Vine | 0.248 ± 0.103 b | 41,495.877 ± 45,318.948 a | 17.148 ± 4.436 b | 0.015 ± 0.014 a | 0.443 ± 0.329 a | 3.105 ± 0.251 a | 35.425 ± 41.661 a | 51.068 ± 8.449 a |
| LPA | Tree | 0.512 ± 0.105 a | 7254.972 ± 1575.539 b | 14.734 ± 2.957 b | 0.005 ± 0.003 b | 0.162 ± 0.056 b | 2.293 ± 0.239 c | 1.918 ± 1.084 c | 45.018 ± 6.056 bc |
| | Shrub | 0.385 ± 0.053 b | 5740.056 ± 3100.166 b | 17.884 ± 2.592 a | 0.009 ± 0.005 ab | 0.577 ± 0.614 a | 2.111 ± 0.476 c | 8.562 ± 8.643 b | 48.891 ± 11.567 ab |
| | Herb | 0.381 ± 0.038 b | 8010.565 ± 5156.448 b | 12.901 ± 1.469 b | 0.006 ± 0.003 b | 0.177 ± 0.076 b | 3.361 ± 0.106 a | 6.29 ± 3.896 bc | 53.133 ± 2.204 a |
| | Vine | 0.253 ± 0.084 c | 27,374.194 ± 27,628.919 a | 12.934 ± 1.397 b | 0.011 ± 0.01 a | 0.201 ± 0.089 b | 2.781 ± 0.781 b | 17.138 ± 11.502 a | 40.67 ± 4.988 c |
| HPA | Tree | 0.491 ± 0.151 a | 6822.454 ± 2559.228 b | 12.926 ± 3.033 b | 0.006 ± 0.003 b | 0.113 ± 0.067 b | 2.466 ± 0.227 c | 1.896 ± 1.17 c | 42.611 ± 9.565 ab |
| | Shrub | 0.381 ± 0.136 b | 7111.725 ± 8816.375 b | 18.764 ± 5.07 a | 0.013 ± 0.013 a | 0.449 ± 0.425 a | 2.697 ± 1.01 bc | 9.002 ± 12.428 a | 47.066 ± 12.162 a |
| | Herb | 0.495 ± 0.092 a | 7959.507 ± 4706.202 b | 13.042 ± 2.754 b | 0.008 ± 0.005 ab | 0.123 ± 0.047 b | 3.868 ± 0.749 a | 4.121 ± 3.28 ab | 49.276 ± 1.217 a |
| | Vine | 0.279 ± 0.109 b | 20,180.6 ± 14,521.167 a | 12.046 ± 1.263 b | 0.008 ± 0.005 ab | 0.179 ± 0.135 b | 3.222 ± 0.825 b | 10.265 ± 7.399 a | 35.589 ± 6.299 b |

(a–c) indicate significant differences in functional traits ($p < 0.05$). Note: leaf dry matter content (LDMC); specific leaf area (SLA); the net photosynthetic rate per unit area ($A_{area}$); stomatal conductance (Gs); transpiration rate (Tr); the nitrogen content of leaves per unit mass ($N_{mass}$); calculation of photosynthetic nitrogen utilization (PNUE); chlorophyll values (SPAD).

### 3.4. Characteristics of Variation in Plant Leaf Traits in Different Gradient Zones

There were interspecific differences in leaf traits among trees, shrubs, lianas, and herbs under different air pollution environment, and the variation of each trait among the 18 garden plants ranged from 8.1% to 153.5% for CAA, 4.1% to 106.5% for LPA, and 2.5% to 138.1% for HPA. The leaf trait with the highest coefficient of variation for both CAA and LPA was Tr, and the leaf trait with the highest coefficient of variation for HPA was PNUE. The coefficients of variation for LDMC, $N_{mass}$, and SPAD were all smaller in the three different air environments. The smallest coefficients of variation for trees in the three different gradients were 15.3%, 10.4%, and 9.2% for $N_{mass}$, respectively. The smallest coefficients of variation for shrubs in CAA and LPA were LDMC, and the largest was Tr. The smallest coefficient of variation for shrubs in HPA was SPAD, and the largest was PNUE. The smallest coefficient of variation for lianas in the three different gradients was SPAD, and the largest coefficient of variation for lianas was PNUE. The minimum value was SPAD in all three groups; the maximum values were SLA in CAA and LPA, and PNUE in HPA. The minimum value of the coefficient of variation for herbs was $N_{mass}$ in CAA, and the minimum value was $A_{area}$ in both contaminated areas (Table 4).

**Table 4.** Characteristics of variation in plant leaf functional traits in different gradient zones.

| Area | Plant-Life | LDMC | SLA | $A_{area}$ | Gs | Tr | $N_{mass}$ | PNUE | SPAD |
|------|-----------|------|------|-----------|------|------|-----------|------|------|
| CAA | Tree | 0.164 | 0.260 | 0.225 | 0.667 | 1.535 | 0.153 | 0.471 | 0.198 |
| | Shrub | 0.125 | 0.403 | 0.140 | 0.615 | 1.129 | 0.283 | 0.789 | 0.250 |
| | Herb | 0.311 | 0.607 | 0.178 | 0.250 | 0.385 | 0.168 | 0.181 | 0.158 |
| | Liane | 0.415 | 1.092 | 0.259 | 0.933 | 0.743 | 0.081 | 1.176 | 0.165 |
| LPA | Tree | 0.206 | 0.217 | 0.201 | 0.620 | 0.344 | 0.104 | 0.565 | 0.135 |
| | Shrub | 0.137 | 0.540 | 0.145 | 0.604 | 1.065 | 0.225 | 1.009 | 0.237 |
| | Herb | 0.099 | 0.644 | 0.114 | 0.417 | 0.431 | 0.032 | 0.619 | 0.041 |
| | Liane | 0.332 | 1.009 | 0.108 | 0.853 | 0.444 | 0.281 | 0.671 | 0.123 |
| HPA | Tree | 0.308 | 0.375 | 0.235 | 0.500 | 0.593 | 0.092 | 0.617 | 0.224 |
| | Shrub | 0.357 | 1.240 | 0.270 | 1.000 | 0.947 | 0.374 | 1.381 | 0.258 |
| | Herb | 0.186 | 0.591 | 0.211 | 0.625 | 0.382 | 0.194 | 0.796 | 0.025 |
| | Liane | 0.391 | 0.720 | 0.105 | 0.625 | 0.754 | 0.256 | 0.721 | 0.177 |

Note: leaf dry matter content (LDMC); specific leaf area (SLA); the net photosynthetic rate per unit area ($A_{area}$); stomatal conductance (Gs); transpiration rate (Tr); the nitrogen content of leaves per unit mass ($N_{mass}$); calculation of photosynthetic nitrogen utilization (PNUE); chlorophyll values (SPAD).

### 3.5. Correlation Analysis between Leaf Functional Traits

#### 3.5.1. Leaf Traits of Plants

As can be seen in Figure 4, among leaf structural traits, LDMC was highly significantly negatively correlated with SLA ($p < 0.01$). Among leaf physiological traits, $N_{mass}$ was highly significantly negatively correlated with $A_{area}$ and significantly negatively correlated with Tr ($p < 0.05$). $A_{area}$ was highly significantly positively correlated with Gs and Tr, and significantly positively correlated with PNUE and SPAD. Gs was highly significantly positively correlated with PNUE, and significantly negatively correlated with SPAD. SPAD was highly significantly negatively correlated with both Tr and PNUE.

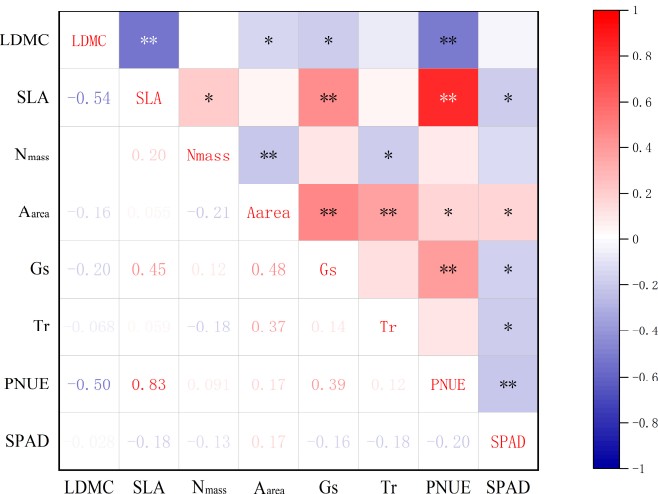

**Figure 4.** Comprehensive correlation analysis of plant leaf functional traits. * ≤ 0.05; ** ≤ 0.01.

Among the structural and physiological traits of leaves, LDMC was significantly negatively correlated with $A_{area}$ and Gs, and highly significantly negatively correlated with PNUE; SLA was significantly positively correlated with $N_{mass}$, highly significantly positively correlated with Gs and PNUE, and significantly negatively correlated with SPAD.

### 3.5.2. Leaf Traits of Plants of Different Life Forms

As can be seen from Figure 5, between leaf structural traits, LDMC and SLA were significantly negatively correlated in tree plant leaves. Between leaf structural and physiological traits, LDMC was highly significantly negatively correlated with $A_{area}$, and SLA was significantly positively correlated with Gs and PNUE. Between leaf physiological traits, $N_{mass}$ was highly significantly positively correlated with Tr, $A_{area}$ was significantly positively correlated with Tr, and Gs was significantly positively correlated with PNUE. SPAD was highly significantly negatively correlated with PNUE and Tr.

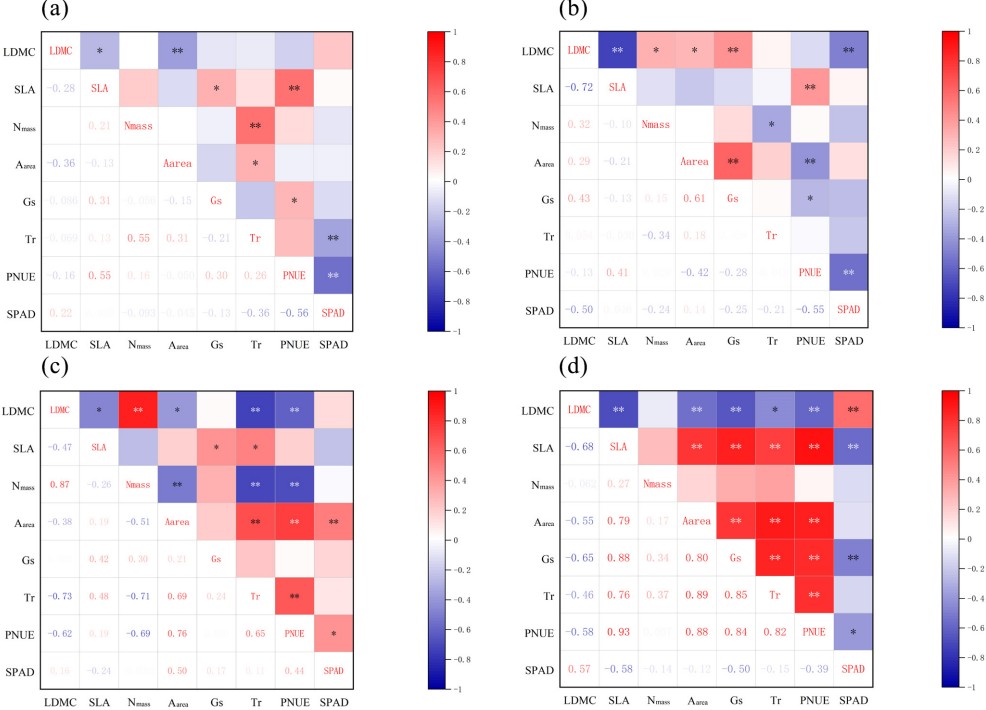

**Figure 5.** Correlation analysis of leaf traits. In the figure, (**a**) trees, (**b**) shrubs, (**c**) lianas, (**d**) herbs. * ≤ 0.05; ** ≤ 0.01.

Between leaf structural traits, shrub plant leaf LDMC was highly significantly negatively correlated with SLA. Between leaf structural traits and physiological traits, LDMC was significantly positively correlated with $N_{mass}$ and $A_{area}$, highly significantly negatively correlated with SPAD, and highly significantly negatively correlated with SLA and PNUE. Between leaf physiological traits, $N_{mass}$ was significantly negatively correlated with Tr, and $A_{area}$ was highly significantly positively correlated with Gs. Gs was highly significant positively correlated with PNUE, Gs was highly significant negatively correlated with PNUE, and PNUE was highly significant negatively correlated with SPAD.

Between the structural traits of leaves, LDMC of lianas was significantly negatively correlated with SLA. between the structural and physiological traits of leaves, LDMC was highly significantly positively correlated with $N_{mass}$, highly significantly negatively correlated with $A_{area}$, highly significantly negatively correlated with Tr and PNUE, and significantly positively correlated with Gs and Tr. Between the physiological traits of leaves, $N_{mass}$ was highly significantly negatively correlated with $A_{area}$, Tr and PNUE were highly significant negatively correlated, $A_{area}$ was highly significant positively correlated with Tr, PNUE and SPAD, Tr was highly significant positively correlated with PNUE, and PNUE was significantly positively correlated with SPAD.

In herbaceous plants, the structural trait LDMC was highly significantly negatively correlated with SLA; between structural shape and physiological traits, LDMC was highly significantly negatively correlated with $A_{area}$, Gs, and PNUE, highly significantly negatively correlated with Tr, and highly significantly positively correlated with SPAD, and SLA was highly significantly positively correlated with $A_{area}$, Gs, Tr, and PNUE, and highly significantly negatively correlated with SPAD; and among the leaf physiological traits, $A_{area}$ was significantly positively correlated with Gs, Tr, and PNUE; Gs was significantly positively correlated with Tr and PNUE and significantly negatively correlated with SPAD; Tr was significantly positively correlated with PNUE; and PNUE was significantly negatively correlated with SPAD.

### 3.6. Fuzzy Affiliation Function Analysis

Among the four life types, the order of adaptability to the environment was trees (0.480) > shrubs (0.418) > herbs (0.374) > lianas (0.367) (Figure 6). Among the trees, the order of adaptability from strong to weak was *Magnolia grandiflora*, *Koelreuteria paniculata*, *Camphora officinarum*, *Osmanthus fragrans*, *Ginkgo biloba*, and *Sapindus saponaria*, respectively. Shrubs were ranked from strongest to weakest in order of adaptability as *Hibiscus mutabilis*, *Pittosporum tobira*, *Loropetalum chinense*, *Viburnum odoratissimum*, *Buxus sinica*, and *Lagerstroemia indica*. The adaptability of herbaceous plants was ranked as *Oxalis corniculata* > *Ophiopogon japonicus* > *Ophiopogon bodinieri*, and that of lianas was ranked as *Parthenocissus tricuspidata* > *Jasminum mesnyi* > *Trachelospermum jasminoides*. The adaptability of plants with different life types in CAA was greater than that in HPA, except for *Buxus sinica*. In HPA, the most adaptable tree was *Magnolia grandiflora* (0.538), and the weakest was *Sapindus Saponaria* (0.359); the most adaptable shrub was *Hibiscus mutabilis* (0.447), and the weakest was *Lagerstroemia indica* (0.339). The strongest herb was *Oxalis corniculata* (0.423), and the weakest was *Ophiopogon bodinieri* (0.316); among lianas, the strongest was *Parthenocissus tricuspidata* (0.405), and the weakest was *Trachelospermum jasminoides* (0.320).

Tree

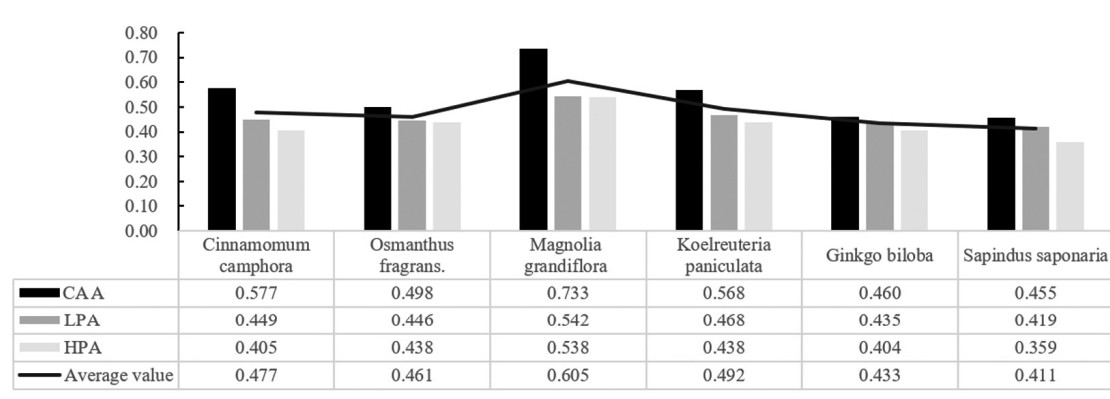

Shrub

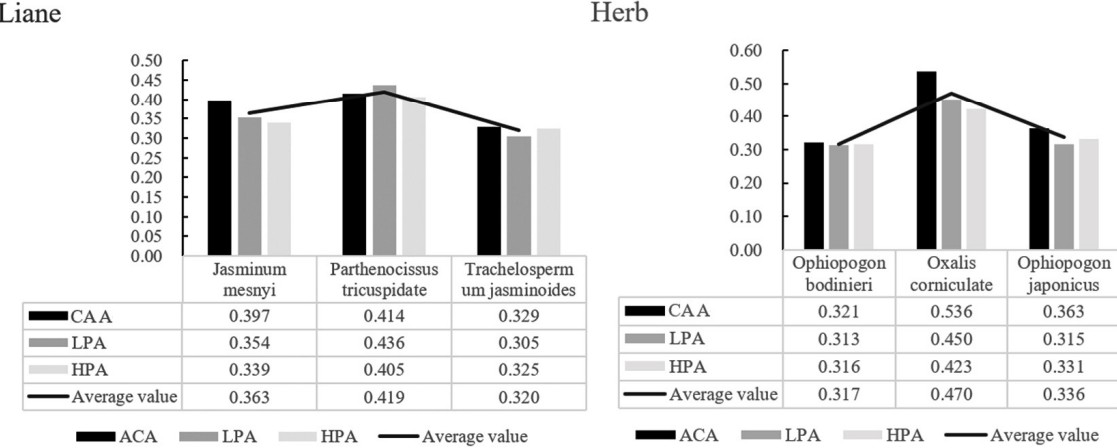

**Figure 6.** Fuzzy affiliation function analysis of plant environmental adaptation.

## 4. Discussion

### 4.1. Leaf Functional Traits in Relation to Air Quality

In this study, we observed a positive correlation between the leaf functional traits specific leaf area (SLA) and leaf nitrogen content ($N_{mass}$), which aligns with the findings of previous research conducted on Jinhua Beishan plants [44]. Furthermore, $N_{mass}$ exhibited a highly significant positive correlation with leaf net assimilation rate ($A_{area}$), which in turn showed positive correlations with stomatal conductance (Gs), transpiration rate (Tr), photosynthetic nitrogen use efficiency (PNUE), and chlorophyll content (SPAD). The positive correlation between SLA and $N_{mass}$ suggests that plants have the ability to modify

their leaf structure to enhance nitrogen accumulation, thereby improving photosynthetic efficiency per unit area and enhancing leaf transpiration rate. This adaptive response aids in mitigating the detrimental effects of air pollution. However, under high-pollution conditions (HPA), we observed a decrease in Tr across all plant life types. This decline may be attributed to the plants' regulation of stomatal [45,46] closure or contraction upon reaching a certain threshold of air pollution. This adaptive mechanism serves to prevent the intrusion of pollutants, consequently reducing transpiration rates.

The size of the SPAD value reflects the chlorophyll content in leaves, and it can be considered an indicator of leaf development. The correlation between $A_{area}$ and SPAD revealed a decrease in both variables under HPA conditions. This decline can be attributed to the inadequate accumulation of organic matter in plant leaves under environmental stress, leading to premature leaf senescence and chlorophyll decomposition, ultimately resulting in a decrease in SPAD values. Du's research presents the same results, with his team finding that chlorophyll a and b are damaged to varying degrees in atmospheric pollution environments, and that total chlorophyll content decreases [47].

The results of redundancy analyses showed that LDMC and $N_{mass}$ were leaf functional traits with consistent and strong correlation with the direction of change of each pollution factor among all plant types, which indicated that LDMC and $N_{mass}$ could be used as good leaf traits to reflect urban air quality. In the face of air pollution stress, plants improved their physiology by increasing LDMC and at the same time regulated the rate of leaf nitrogen building to potentially affect their photosynthetic utilization efficiency and chlorophyll value, improving their adaptability to air pollution environments. This is consistent with the findings of Juanxia Li [48] and Jiyou Zhu [39].

### 4.2. Relationships between Leaf Traits of Different Life Types

Differences in functional traits among species serve as the foundation for species coexistence in natural ecosystems, while intraspecific trait variation also contributes significantly to species coexistence and distribution [49]. Only by considering both intraspecific and interspecific trait variations can we accurately assess the response of species to environmental changes and resource competition during community development [50]. Leaf dry matter content (LDMC) serves as a critical indicator of nutrient resource conservation in plants [51]. High LDMC values indicate the ability of plants to thrive in resource-limited environments, enabling them to develop growth responses and maintain nutrient conservation efficiency [52]. In this study, the mean LDMC value for trees, shrubs, lianas, and herbs in Suzhou were found to be 0.479 $g \cdot g^{-1}$, 0.380 $g \cdot g^{-1}$, 0.371 $g \cdot g^{-1}$, and 0.26 $g \cdot g^{-1}$, respectively. The higher LDMC values observed in trees and shrubs suggest their enhanced resistance to physical damage. Moreover, the significant increase in LDMC among all plant types in areas with higher pollution levels reflects the leaf's efficient nutrient stabilization mechanism, enabling survival in more polluted air environments.

Research has demonstrated that plants with lower specific leaf area (SLA) values exhibit efficient nutrient conservation, while those with higher SLA values are better at capturing light and have relatively higher growth rates [53]. We investigated the variations in SLA among plant life forms in different air pollution gradients. The results show significant differences in SLA between herbaceous plants and other plant types across the air quality gradients. Additionally, SLA values for all plant types were negatively correlated with air pollution levels. Trees and shrubs exhibited lower SLA values compared to herbs and lianas, which occupy the middle and upper layers of the community and are influenced by factors such as strong light. These tree and shrub species allocate more biomass and nitrogen to build cell walls, enabling them to accumulate photosynthesis products necessary for overwintering and enhancing leaf toughness [54]. In contrast, herbaceous plants tend to allocate more nitrogen to thylakoids and carboxylases involved in respiration, resulting in higher photosynthetic capacity, as evidenced by their larger leaf area [55], lower leaf thickness, and higher SLA values [56].

We aimed to investigate the variations in $A_{area}$, Tr, and Gs among different plant life forms. The mean $A_{area}$ values were 14.713 μmolm$^{-2} \cdot$s$^{-1}$ for trees, 19.116 μmolm$^{-2} \cdot$s$^{-1}$ for shrubs, 14.903 μmolm$^{-2} \cdot$s$^{-1}$ for lianas, and 14.043 μmolm$^{-2} \cdot$s$^{-1}$ for herbs. The $A_{area}$ of shrubs was significantly higher than that of other plant types, indicating their greater ability to absorb $CO_2$ and nitrogen from the air and use them for photosynthesis, ultimately improving plant growth efficiency and productivity. Furthermore, $A_{area}$ was significantly correlated with the Tr and Gs of different plant types, which is consistent with the findings of Shao et al.'s study on the main greening tree species in Shanghai [57].

We also examined the SLA, $N_{mass}$, and PNUE of different plant types. The results reveal that herbaceous plants had significantly higher SLA, $N_{mass}$, and PNUE values compared to trees, shrubs, and lianas. This finding is similar to the results of Song He et al.'s study on the leaf economic spectrum of Beijing Botanical Gardens, which suggests that the specific combination of structural and physiological traits of herbaceous plants, such as thin leaf blades and high photosynthetic capacity, gives them a competitive advantage in their habitat [58].

### 4.3. Analysis of Plants in the Leaf Economic Spectrum

The leaf economic spectrum represents a quantitative manifestation of the variation in plant leaf functional traits, as assessed through a series of trait indicators (Figure 7). In this study, we observed significantly higher values of SLA, $N_{mass}$, and PNUE in herbs and lianas compared to shrubs and trees, with the lowest values found in trees. This suggests that herbs and lianas occupy the "fast investment-return" end of the leaf economic spectrum, characterized by high nutrient concentrations, rapid photosynthetic rates, fast respiration rates, and short lifespans. On the other hand, shrubs and trees occupy the "slow investment-return" end of the spectrum, characterized by low nutrient concentrations, high photosynthetic rates, slow respiration rates, and long lifespans.

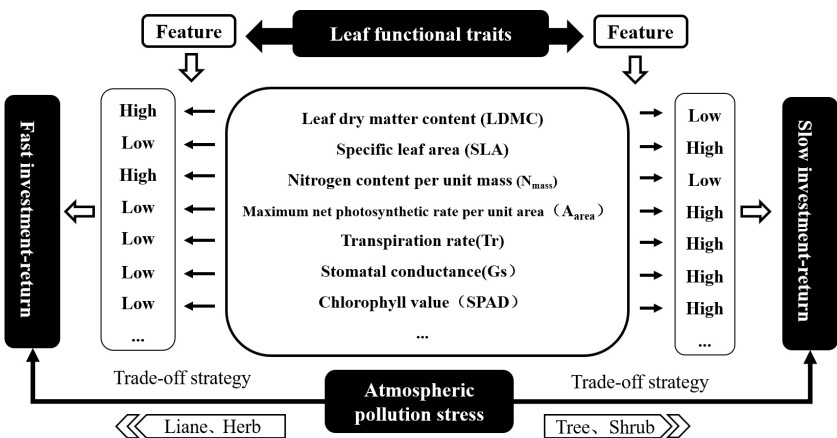

**Figure 7.** Leaf economic spectrum under the stress of atmospheric pollution [59–61].

However, the position of different life types on the leaf economic spectrum is also shifted under different atmospheric pollution environments. We observed that trees under LPA and HPA were biased towards the "fast investment-return" end of the spectrum, whereas plants in CAA were biased towards the "slow investment-return end"; however, we did not observe any significant positional differences for shrubs. The largest deviation in the leaf economic spectrum was for lianas, with very significant shifts to both ends of the spectrum for both plants in CAA and HPA; herbs were second only to lianas, but there was not much difference between the shifts for plants in the CAA and LPA environments. Overall, plants in CAA were more oriented towards the "slow investment-return end", while plants in LPA and HPA were more oriented towards the "fast investment-return" end, which confirms that plants will make different levels of resource This confirms that plants make different levels of resource trade-offs in air pollution environments.

*4.4. Research Limitations*

In this study, the relationship between leaf economic spectral traits and the effect of air pollution on 18 plant species in Suzhou City was analyzed only in a single season of a single year, while environmental factors such as temperature, wind speed, potential evapotranspiration, and water [62,63] have seasonal differences and may lead to variations in the trade-off strategies of the plants and cause bias in the experimental results. In addition, this may also introduce a bias in the relationship between leaves to air pollutants and the overall plant response to the environment are different. In future studies, we can continue to expand the research level by combining the functional traits of the whole plant, or set up experimental sample plots by considering the effects of leaf litter and roots on air pollution, and conduct research from a community perspective by combining leaf area index, plant height and crown size, and try to monitor the experiments for many years in order to ensure a more comprehensive experiment.

**5. Conclusions**

The results of this study on garden plants in Suzhou City show correlations among leaf economic spectrum traits that are consistent with global patterns. In this study, we investigated the response of plant leaf traits under urban air pollution, enriched the small-scale study of leaf economic spectrum theory, and proposed a plant-screening model under different air pollution environments so as to make the plant configuration more in line with the objective quantitative standards, and to provide a model reference for the construction of scientific, reasonable and ecologically balanced urban parks and green spaces under the air pollution environment.

**Author Contributions:** Z.Y., experimentation, data analysis, writing and editing; X.Z. and F.G., review and editing; Y.Q., project administration; Y.L., experimentation. All authors have read and agreed to the published version of the manuscript.

**Funding:** This research was funded by "Heilongjiang academy of sciences shuangtiyanzhen project (grant NO. STYZ2022ZR01)"; "Jiangsu postgraduate research and innovation programme (grant No. KYCX22_3310)"; and "14th Five-Year Plan" Jiangsu Province Key Discipline Construction Project Funding (Landscape Architecture) (grant No. 082240006/002/002).

**Data Availability Statement:** Data are contained within the article.

**Conflicts of Interest:** The authors declare no conflict of interest.

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
