# Peer review of "Response of Common Garden Plant Leaf Traits to Air Pollution in Urban Parks of Suzhou City (China)"

_forests, doi:10.3390/f14112253_

Round 1

Reviewer 1 Report

Comments and Suggestions for Authors

The authors studied physiological and structural response of trees, shrubs herbs and lianas under different air pollution locations. The topic is sound and can be interesting for scientistis. Introduction, Material and methods, Results, Discussion and Conclusion should be improved. I recommend major revision and gave some remarks, comments and suggestions in the Annotated Manuscript (pdf version), so that the authors can improve this paper for this journal.

Comments on the Quality of English Language

The authors studied physiological and structural response of trees, shrubs herbs and lianas under different air pollution locations. The topic is sound and can be interesting for scientistis. Introduction, Material and methods, Results, Discussion and Conclusion should be improved. I recommend major revision and gave some remarks, comments and suggestions in the Annotated Manuscript (pdf version), so that the authors can improve this paper for this journal.

Reviewer 2 Report

Comments and Suggestions for Authors

A few critical comments and remarks on the article by Yang et al. “Response of common garden plants to air pollution in urban parks of Suzhou City”.

The authors must correctly specify the Latin names of the plants. First, you must provide an accepted biological name, not synonyms:

Camphora officinarum Nees - this name is a synonym of Cinnamomum camphora (L.) J.Pres (accepted Latin name).

Second, the first mention of the species' Latin name in the text should be given in full. Indicate each plant's full Latin name in Table 1.

The type of atmospheric air pollution in the research region is unknown. What are the primary sources of pollution? Enterprises? What type? Power plants? Oil refineries? Steel industries or chemical plants? Figure 1 should show where they are located. Alternatively, is high traffic the primary cause of pollution?

What is the total number of pollutants emitted into the city atmosphere?

The wind rose should be shown in Figure 1. Only then will it be possible to comprehend the reasoning behind choosing sample locations in relation to air pollution sources while taking the level of pollution into consideration (HPA > LPA > CAA).

The sampling procedure is described very formally.

“…sampling was conducted in autumn (early October to late November)…” (Lines 145-146). How many times was sampling performed throughout this period?

When was the sample taken during the day? Morning, noon, or evening?

How many leaf samples from each plant species were collected for analysis?

“…to ensure that the plants were of similar age…” (Lines 146-147). Provide information on the age of the trees that have been chosen for examination (for each species). Were they single trees or did tree stands serve as the sampling site?

“... to investigate the correlation between leaf traits of different landscape plants and thermal environmental factors…” (Lines 190-192). Correlation with thermal environmental factors? In the article, this information is absent. Include these data in the Results section.

Figure 7. Between "big" and "high", what is the difference?

Subsection 4.3. “…the "fast investment-return"…, …the slow investment-return"…”. What new have the authors discovered? This is a well-known fact. This is the evolutionary difference in growth and development tactics between annual and perennial plants.

Add research limitations to the Discussion.

Round 2

Reviewer 2 Report

Comments and Suggestions for Authors

I must acknowledge that the authors have made an impressive effort in taking into account all the suggestions I made in my revission. In its current state I consider that the text has been reached the publishing level and can be accepted.